# COVID-19 and Adult Acute Leukemia: Our Knowledge in Progress

**DOI:** 10.3390/cancers14153711

**Published:** 2022-07-29

**Authors:** Franziska Modemann, Susanne Ghandili, Stefan Schmiedel, Katja Weisel, Carsten Bokemeyer, Walter Fiedler

**Affiliations:** 1Department of Oncology, Hematology and Bone Marrow Transplantation with Section Pneumology, University Cancer Center Hamburg, University Medical Center Hamburg-Eppendorf, Martinistraße 52, 20246 Hamburg, Germany; s.ghandili@uke.de (S.G.); k.weisel@uke.de (K.W.); c.bokemeyer@uke.de (C.B.); fiedler@uke.de (W.F.); 2Mildred Scheel Cancer Career Center, University Cancer Center Hamburg, University Medical Center Hamburg-Eppendorf, Martinistraße 52, 20246 Hamburg, Germany; 3The I. Department of Internal Medicine, Division of Infectious Diseases, University Medical Center Hamburg-Eppendorf, Martinistraße 52, 20246 Hamburg, Germany; s.schmiedel@uke.de

**Keywords:** adult acute leukemia, COVID-19, SARS-CoV-2

## Abstract

**Simple Summary:**

We herein report a review of the current literature on adult patients with acute leukemia (AL) infected with SARS-CoV-2. SARS-CoV-2-associated mortality ranges from 20–52% in adult patients with AL, and patients with acute myeloid leukemia have a particularly high COVID-19-related mortality. Of note, most of the available data relate to the pre-vaccination era and to variants before Omicron. Based on expert opinions, the current recommendations suggest delaying systemic AL treatment in SARS-CoV-2-positive patients until SARS-CoV-2 negativity, if immediate AL treatment is not required. It is recommended to offer vaccination to all AL patients, and seronegative patients should additionally receive prophylactic administration of anti-SARS-CoV-2 monoclonal antibodies. Patients with AL infected with SARS-CoV-2 should be treated early with antiviral therapy to prevent disease progression and to enable the rapid elimination of the virus.

**Abstract:**

The majority of publications regarding SARS-CoV-2 infections in adult patients with acute leukemia (AL) refer to hematological patients in general and are not focused on acute myeloid leukemia (AML) or acute lymphoblastic leukemia (ALL). We herein report a review of the current literature on adult AL patients infected with SARS-CoV-2. Overall, SARS-CoV-2-associated mortality ranges from 20–52% in patients with adult AL. AML patients have a particularly high COVID-19-related mortality. Of note, most of the available data relate to the pre-vaccination era and to variants before Omicron. The impact of COVID-19 infections on AL treatment is rarely reported. Based on the few studies available, treatment delay does not appear to be associated with an increased risk of relapse, whereas therapy discontinuation was associated with worse outcomes in AML patients. Therefore, the current recommendations suggest delaying systemic AL treatment in SARS-CoV-2-positive patients until SARS-CoV-2 negativity, if immediate AL treatment is not required. It is recommended to offer vaccination to all AL patients; the reported antibody responses are around 80–96%. Seronegative patients should additionally receive prophylactic administration of anti-SARS-CoV-2 monoclonal antibodies. Patients with AL infected with SARS-CoV-2 should be treated early with antiviral therapy to prevent disease progression and enable the rapid elimination of the virus.

## 1. Introduction

In late 2019, the first cases of pneumonia caused by the novel severe acute respiratory syndrome coronavirus 2 (SARS-CoV-2) were described in Wuhan, China [1]. Since then, the virus has rapidly spread worldwide, causing more than 525 million confirmed cases of SARS-CoV-2 infection with more than 6.2 million deaths [2].

The course of the disease—called corona virus infection disease 2019 (COVID-19) by the World Health Organization (WHO)—is nonspecific and can vary widely. The main route of infection is droplet and/or aerosol transmission [3]. Therefore, spatial distancing, contact restriction, wearing a protective medical mask, and hygiene measures were generally recommended. Most infections in the general population are mild; around 85% in the pre-vaccination era had mild symptoms such as fever, cough, fatigue, and headache; up to 10% had critical courses, and the mortality rate was around 5–7% [4]. The clinical symptoms are not unique and can range from fever, dry cough, and malaise to life-threatening pneumonia with acute respiratory distress syndrome disseminated intravascular coagulopathy, and associated thrombotic events or neurologic dysfunction and cardiomyopathy, multiorgan failure, and death [5,6]. The average incubation period of SARS-CoV-2 is about five days, but can last two weeks until the infected person becomes ill [7]. However, the infected person may be contagious even for several days before the onset of any symptoms [8]. This often makes it difficult to trace contacts to break chains of infection. At the same time, it is also becoming difficult in hospitals to isolate patients to properly prevent infections. Furthermore, during the pandemic, different SARS-CoV-2 variants were identified to carry the risk of different virulence, dissemination in the organism, alteration of the disease pattern, or increased mortality [9,10].

By the end of 2020, the US Food and Drug Administration (FDA) authorized the first- and second-in-class SARS-CoV-2 messenger ribonucleic acid (mRNA) vaccines BNT162b2 and mRNA1273 for emergency use [11,12]. The European Medicines Agency (EMA) followed shortly after with the additional approval of the first-in-class anti-SARS-CoV-2 vector vaccine AZD1222 [13,14,15]. Until now, over 10 billion vaccine doses have been administered worldwide [2]. However, the different SARS-CoV-2 variants also play a role in vaccine-induced immunity, with some subtypes in which the reduced effectiveness of protective vaccinations could be observed [16,17].

In general, the risk of pneumonia from respiratory viruses is higher in cancer patients, which also applies to infections with SARS-CoV-2 [18,19,20,21]. A large Chinese pre-vaccination study identified cancer as the highest individual risk factor for severe events during COVID-19 in a logistic regression model (OR: 5.4, *p* = 0.003) [19]. Moreover, cancer patients had significantly higher rates of severe COVID-19 courses (including admission to the intensive care unit, invasive ventilation, or death) compared with non-cancer patients (*p* = 0.0003) [19]. Recent meta-analyses reported high COVID-19-associated mortality rates in cancer patients ranging between 17 and 26% [22,23,24]. Patients with hematological malignancies and particularly patients with acute leukemia (AL) during induction chemotherapy are at high risk of developing infection complications due to profound and prolonged neutropenia [25,26]. Our knowledge of the disease course and management of patients with AL and COVID-19 is constantly changing. However, only a few comprehensive reviews provide an overview of the clinical features, prevention, and treatment in this particular population.

Most of the available data on the impact of COVID-19 in hematological patients include various hematologic malignancies and do not focus on acute myeloid leukemia (AML) or acute lymphoblastic leukemia (ALL). For this reason, the transfer of previous findings in hematological patients to AL patients is restricted. Treating patients with AL and concomitant COVID-19 challenges hematologists worldwide and requires interdisciplinary cooperation between hematologists and infectiologists. On the one hand, newly diagnosed AL requires an urgent need for chemotherapy without long delays. On the other hand, intensive therapy might increase the risk of severe courses of COVID-19. Therefore, the systemic treatment of AL has to be considered carefully.

Nowadays, with targeted therapies, e.g., FLT3-inhibitors in AML [27], tyrosine-kinase inhibitors in ALL [28], or antibody/immunotherapies such as blinatumomab [29] or gemtuzumab/ozogamicin [30], more precise and possibly less toxic or less immunosuppressive therapies for treating AL patients are available and might impact the decision for choosing individual regimens in AL patients with COVID-19.

We herein report a review of the current literature, regarding the clinical courses, the impact of COVID-19 on the treatment of AL, the reported response to vaccination, the application of COVID-19 antibodies, and the general recommendations for patients with AL and concomitant SARS-CoV-2 infection. This review does not include data or recommendations on allogeneic stem cell transplantation.

## 2. Clinical Courses

Overall, across all COVID-19 waves, the SARS-CoV-2-associated mortality ranged from 20–52% in patients with AL [31,32,33]. However, no comprehensive data are yet available for the Omicron variant.

One of the largest multi-center trials investigating the impact of COVID-19 in adult AML patients was recently published by the European Hematology Association [31]. The EPICOVIDEHA registry included 388 adult AML patients with a COVID-19 diagnosis between February 2020 and October 2021. Severe and critical courses of COVID-19 occurred in 41% and 21%, respectively. Death due to COVID-19 was reported in 20% of patients. Ongoing or recent AML treatment (defined as less than three months before SARS-CoV-2 infection) was associated with higher mortality rates due to COVID-19, as well as active leukemic disease, older age, and treatment discontinuation. Significantly lower overall survival was observed in patients diagnosed between January 2020 and August 2020 (first wave), compared to those after September 2020 (41% vs. 25%, *p* < 0.0001) [31]. This is the only more extensive data set that allows a comparison between the different SARS-CoV-2 waves. Our own experience from our department—treating around 90 newly diagnosed AL patients per year—underlines the results reported by the EPICOVIDEHA trial by observing no cases of severe or critical COVID-19 courses in AL patients since Omicron appeared. Previous results also reported by the EPICOVIDEHA registry trial from the pre-vaccination era regarding patients with hematological malignancies showed a high overall mortality rate of 40% in the subset of adult AML and 26% in adult ALL patients [34]. In the entire cohort of patients with hematological malignancies, AML and high-risk myelodysplastic syndrome (MDS) displayed the highest overall mortality, and AML was the only independent risk factor for mortality in the multivariate analysis [34]. Another two large observational studies reported by the PETHEMA group from the first SARS-CoV-2 wave included 108 and 117 adult AML patients, showing a comparable high mortality rate of 43.5% and 48%, respectively [35,36]. Interestingly, the mortality rates reported from four Latin American countries with limited medical resources did not differ, with a mortality rate of 37% in 83 patients with AL [33]. However, Mitrovic et al. reported a lower mortality rate of 29% in 51 patients with AL from the pre-vaccination era in Serbia, with worse overall survival in patients with newly diagnosed leukemia and those with profound neutropenia [32]. A large proportion of other published data sets include hematological patients in general [37,38,39,40,41]. García-Suárez et al. reported on one of the largest sub-cohorts of AL patients, showing 48% severe courses of COVID-19 and 26% critical courses in 61 patients with AML, leading to an overall mortality rate of 44% [38]. This region- or country-specific high mortality rates are paralleled by an early global data set of the American Society of Hematology (ASH) Research Collaborative COVID-19 Registry for Hematology, which describes a mortality rate of 33% in 80 patients with AL and concomitant SARS-CoV-2 infection [42]. Moreover, two important publications by Italian and Spanish multicenter registries reported the subgroup of adult ALL patients infected with SARS-CoV-2, describing an overall mortality rate between 11% and 33% [43,44].

Most of the further literature explicitly describing the clinical course of COVID-19 in adult AL patients is based on small case series and single case reports [40,45,46,47,48,49].

General risk factors for developing severe and critical courses of COVID-19 are summarized by the European Conference on Infections in Leukemia 9 (ECIL 9) statement: older age, cardiovascular and metabolic comorbidities, and an active or uncontrolled malignancy represent the main risk factors for mortality in AL patients who are infected with SARS-CoV-2 [50].

A summary of the most important literature is presented in Table 1 and Table 2.

## 3. Impact of COVID-19 on AL Treatment

Publications on the impact of COVID-19 on AL treatment and recommendations for treatment modifications in patients with AL are scarce or mostly based on expert opinion. This gap in knowledge makes it difficult to establish the best therapeutic strategies for patients with AL and concomitant SARS-CoV-2 infection. The two main questions arising regarding the therapy of adult AL patients in the setting of COVID-19 are the following: (1)should AL treatment be delayed or discontinued in the case of SARS-CoV-2 positivity?(2)can intensive AL treatment schedules still be considered during the COVID-19 pandemic?

In the most recently published cohort of 388 adult AML patients, chemotherapeutic schedules were modified in 45% of patients. Of those, therapy was delayed in 39% and treatment was discontinued in 61% [31]. Interestingly, Marchesi et al. showed that a treatment delay was protective in a multivariate analysis for the clinical course of COVID-19, whereas therapy discontinuation was associated with worse outcomes. These results were also applicable to patients with newly diagnosed AML in whom a treatment delay was possible [31]. The Latin American study investigating 83 patients with AL reported therapy modifications in 49% of patients; the main modification was a delay of treatment, followed by a change in therapy scheme and a dose reduction (no data available about discontinuation). None of the treatment modifications were significantly associated with an increased risk of relapse [33]. At the beginning of the COVID-19 pandemic in 2020, Zeidan et al. recommended to post-pone all AL therapies for 10–14 days in SARS-CoV-2-infected patients, if the immediate treatment of AL was not deemed necessary [51]. This recommendation still applies. If possible, induction treatment should be delayed in the case of SARS-CoV-2 positivity until PCR negativity, according to the current ASH and ECIL 9 recommendations [50,52]. Of note, the time of SARS-CoV-2 clearance might be prolonged in AL patients for up to 82 days [36,45,53]. Furthermore, despite the COVID-19 pandemic, consolidation therapy with high-dose cytarabine should be offered to patients in complete remission. High-dose cytarabine should be reduced to 1.5 g/m^2^ and a reduction in the number of consolidation cycles should be considered [51,52,54].

Most reported data sets describing differences in the outcome of AL patients treated with intensive or non-intensive treatment schedules are based on small case series. Remarkably, treatment with demethylating agents before COVID-19 in patients with hematological malignancies displayed the highest mortality rate with 59% in the first published EPICOVIDEHA registry trial [34]. However, these results were not confirmed by the most recent study published by Marchesi et al., who did not observe significant differences between therapeutic approaches of different intensity, and demethylating agents in particular [31]. Nevertheless, these results contrast the observations of Garcia et al. and the Asociación Madrileña de Hematología y Hemoterapia group, reporting a significantly lower mortality rate in 32 patients with AML or MDS receiving hypomethylating agents before COVID-19 [38].

According to the current recommendations of the ASH, the European Society for Blood and Marrow Transplantation (ESBMT), and Zeidan et al., intensive chemotherapy should be offered to AML patients who are considered eligible for intensive chemotherapy [51,52,55]. However, to avoid hospitalization and to reduce transfusion frequency in the case of local outbreaks or a shortage of beds and blood supplies, low-dose therapy with azacytidine in combination with venetoclax may be considered as an alternative, especially in those patients with reduced performance status [38,45,46,52,56,57].

Large data sets are missing for adult ALL patients with COVID-19 due to the low ALL incidence in adults. The ASH recommends delaying systemic ALL treatment. Since the detection of SARS-CoV-2 can be prolonged for up to 91 days in ALL patients, it remains unclear how to proceed with these patients [43]. However, intrathecal therapy should not be delayed in the case of central nervous system symptoms [58]. Since corticosteroids are essential components of ALL treatment, it is highly recommended by the ESBMT to proceed with a standard dosage of corticosteroids in ALL patients, even if there might be a potential risk for provoking viral rebound [55]. Furthermore, the administration of peg-asparaginase as an essential compound in ALL treatments is challenging due to the increased risk of thrombotic events caused by COVID-19. Nevertheless, peg-asparaginase should be administered under the intensive monitoring of coagulation parameters in SARS-CoV-2-infected ALL patients according to the ESBMT recommendations [55]. For Philadelphia chromosome-positive ALL patients, it is recommended to switch the therapeutic approach to tyrosine-kinase-inhibitor-based treatment in combination with steroids rather than multi-agent chemotherapy [58,59]. This approach has been reported as feasible by the CAMPUS ALL Network [43]. The use of rituximab during consolidation therapy is controversial and no clear recommendations are available [58]. It is recommended to withhold the treatment of patients during maintenance therapy until symptoms are resolved for at least two weeks [58].

Whether profound and prolonged neutropenia is a risk factor for a high mortality rate in AL patients with COVID-19 is still controversial and remains unanswered. In a recent US study, neutropenia seven days before and 28 days after SARS-CoV-2 infection was identified as an independent risk factor for higher mortality [60]. In cancer patients in general, severe neutropenia was also associated with a worse outcome [61]. However, in the large study of 388 adult AML patients by Marchesi et al., neutropenia was not identified in the multivariate analysis as an independent risk factor in COVID-19 [31]. Nevertheless, in the current statement of the ASH and by Zeidan and colleagues, experts recommend strongly considering the use of growth factors to reduce the duration of neutropenia in patients who are not suffering from severe COVID-19 [51,52]. Since there is a rare but potential risk of aggravating COVID-19-related inflammatory pulmonary injury in patients with a severe and critical course of COVID-19 by applying granulocyte stimulating factor, the administration of growth factors is not recommended in this patient cohort [50,51].

## 4. General Recommendations

Summarizing the currently available literature, in patients with AL it is recommended to:continue environmental precautions such as wearing a face mask, social distancing, reducing, e.g., visitor restriction, ventilation of rooms, patient isolation, and following general hygiene measures [32,50,62,63,64]before starting treatment, screen for COVID-19 as close to treatment start as possible [51,52,55,58,62,65]universally screen of all visitors [62]test for COVID-19 if a patient has COVID-19-specific symptoms regardless of the time point of the last negative COVID-19 test [32,52,62]use approved disinfectants for regularly decontaminating surfaces in hospitals [62,66,67]minimalize hospital stay by discharging early [62]shift as many aspects of care into the outpatient setting as possible [50,51,52,54,55,58,62]

## 5. Vaccination and Treatment of COVID-19

The global impact of the current COVID-19 pandemic has led to the impressively rapid development of multiple SARS-CoV-2-directed vaccines, which are each highly effective in healthy populations. In all AL patients, SARS-CoV-2 vaccination is generally recommended by several hematological societies [50,52,58]. In contrast to most B-cell malignancies after complete vaccination with mRNA vaccines, high serological responses of 80% to 96% were observed in the subsets of AML and ALL patients after complete vaccination with mRNA vaccines [68,69,70,71]. Up to 96% of seroconversion has been observed in elderly AML patients treated with demethylating agents with or without venetoclax [70]. As previously described, the serological vaccination response is impaired in patients who received CD20-directed treatment, which is used in most ALL patients [68].

In addition to vaccination, the intramuscular application of a single dose of the two SARS-CoV-2 neutralizing spike antibodies tixagevimab and cilgavimab should be considered for pre-exposition prophylaxis in AL patients [72]. Therefore, all AL patients should be tested for their antibody levels before AL-specific treatment, irrespectively of their vaccination status. The recently published results of the ongoing phase III PROVENT trial, which included a comparably small subset of 197 immunosuppressed patients, showed an overall risk reduction of 83% for COVID-19-related hospitalization when comparing tixagevimab and cilgavimab to a placebo [73]. Other neutralizing spike antibodies showing promising results regarding the pre-exposition prophylaxis of COVID-19 are casirivimab and imdevimab, and the monoclonal antibody sotrovimab [74]. However, casirivimab, imdevimab, and sotrovimab showed less efficacy in vitro against the currently predominant SARS-CoV-2 variant Omicron and also appeared to be less effective in patients than the other neutralizing spike antibodies in reducing the time to recovery from the SARS-CoV-2 Omicron variant [75,76]. Therefore, the therapeutic impact of anti-spike-protein antibodies for the prevention and treatment of currently circulating SARS-CoV-2 variants (BA.2, BA.4, BA.5) remains unclear. An exception is bebtelovimab, a monoclonal antibody that showed efficacy against the Omicron variant and has now achieved an emergency use authorization by the FDA for the treatment of mild-to-moderate COVID-19 in non-hospitalized patients who are at high risk of progression to severe COVID-19, for whom alternative COVID-19 treatment options are not accessible or clinically appropriate based on the results of the BLAZE-4 trial (NCT04634409). However, up to now it is unknown if patients with acute leukemia were included in the BLAZE-4 trial [77]. In patients with an increased risk of severe courses of COVID-19, several antivirals are now available to prevent severe or critical courses. The orally administered combination of nirmatrelvir/ritonavir demonstrated a reduction in COVID-19-associated hospitalization and mortality compared to the placebo from 7% to 0.8% in non-hospitalized patients with COVID-19 and without additional oxygen requirements, but with at least one risk factor for a severe course of COVID-19, including patients with immunosuppression [78]. Drug interactions, particularly with venetoclax due to its metabolism via CYP3A4 enzymes, must be considered when treating AL patients with antiviral compounds. Moreover, early treatment with molnupiravir therapy also reduced the risk of hospitalization or death in non-hospitalized adults with laboratory-confirmed SARS-CoV-2 infection and at least one risk factor for severe COVID-19, including cancer patients, from 10% to 7% compared to the placebo [79]. However, both trials reported on the risk reduction in an unvaccinated study population. Moreover, early treatment with remdesivir was associated with a reduction in hospitalization or mortality from 5% to 0.7% among non-hospitalized adults who were at high risk for severe COVID-19, compared to the placebo [80]. Furthermore, an observational trial including 313 patients with hematological malignancies showed a reduced mortality rate in patients treated with remdesivir [81]. The results of reducing severe courses in COVID-19 by administering antivirals cannot generally be transferred to COVID-19 vaccine breakthrough infections since the results are mainly based on cases from the pre-vaccination era. The use of other therapeutic agents with antiviral efficacy such as (hydroxy-) chloroquine, lopinavir/ritonavir and azithromycin have not been studied in clinical trials, and these compounds should not be used. This also applies to ivermectin, arbidol, or favipiravir [50,82,83].

In addition to antivirals, monoclonal anti-SARS-CoV-2 antibodies (SARS-CoV-2-moab) are available for treating SARS-CoV-2-infected patients, such as sotrovimab, casirivimab/imdevimab, bamlanivimab/etesevimab, bebtelovimab or regdanvimab. There is very limited data on the use of these antibodies in patients with AL. The choice of the antibody for treating COVID-19 should be based on national approval, local availability, and the current, local circulating SARS-CoV-2 variants, since not all antibodies could show efficacy against all SARS-CoV-2 variants [50,75,76]. Sotrovimab showed the highest efficacy against the Omicron variant amongst all SARS-CoV-2-moab and reduced the hospitalization and death rates in patients at risk for developing a severe COVID-19 course [84,85]. Unfortunately, there is a lack of information regarding the efficacy in patients with AL. Nevertheless, SARS-CoV-2-moab have low side effects [86,87] and the risk for interactions with AL treatment medication is low. However, judging from the laboratory data, all available monoclonal anti-SARS-CoV-2 antibodies seem to have reduced or no activity against the currently circulating Omicron virus variants [74,75].

Another option to prevent developing severe SARS-CoV-2 courses is the treatment with convalescent plasma. An early administration <72 h after SARS-CoV-2 infection and high-titer convalescent plasma showed a reduced progression rate to severe COVID-19, but in the meta-analysis, higher recovery rates or improved survival were not observed [88,89]. Nevertheless, in patients with hematological malignancies, a statistically significant benefit for clinical recovery and SARS-CoV-2-associated mortality was observed [90]. As SARS-CoV-2-moab, convalescent plasma has low side effects and low risk for drug interactions, but after the approval of the SARS-CoV-2-moabs, this therapy has clearly lost its importance due to the residual risk of infection transmission from the donor.

The main cause of mortality in SARS-CoV-2-infected patients is the development of hyper-inflammation syndrome [91]. There are different therapeutic strategies to treat COVID-19 patients who develop hyper-inflammation. Steroid therapy with dexamethasone 6 mg for 10 days demonstrated a 3% reduction in mortality in the large RECOVERY trial compared to the usage of usual care [92]. Of note, steroids should only be administered in those patients who enter the hyper-inflammatory phase and not in patients in the early viral phase who do not need oxygen therapy, since adverse effects have been shown in these patients [91]. Higher dosages of dexamethasone than 6 mg are not recommended [93]. In addition to steroids, therapy with anti-interleukin-1 (IL-1) and anti-interleukin-6 (IL-6) monoclonal antibodies has also been investigated for patients with hyper-inflammatory syndrome. Anti-IL-6 treatment showed a benefit in clinical recovery and reduced the rate of disease progression and mortality in patients under oxygen therapy [94,95,96]. For anti-IL-1 treatment, only one study is available, showing the effectiveness of anti-IL-1 administration in patients with high a inflammatory state [97]. Studies on patients with AL are missing.

Treatment recommendations in the case of infection with SARS-CoV-2 in AL patients are dynamically changing. The last recommendations were published by the ECIL in April 2022, suggesting the following approaches for patients with AL based on the available data described above [50]:in AL patients with mild COVID-19 disease, SARS-CoV-2-moab, high-titer convalescent plasma for seronegative patients, molnupiravir, nirmatrelvir/ritonavir, and/or remdesivir are recommended. Dexamethasone is not recommended [50]in AL patients with moderate or severe COVID-19 disease, remdesivir, dexamethasone, anti-IL-1 antibody, and anti-IL-6 antibody are recommended. If patients are seronegative, SARS-CoV-2-moab or convalescent plasma is additionally suggested. It is not recommended to discontinue or adjust any AL-specific immunosuppressive therapy [50]in AL patients with critical COVID-19 disease, dexamethasone and, if necessary, anti-IL-6 antibody treatment is recommended. If patients are seronegative, SARS-CoV-2-moab and/or high titer convalescent plasma are additionally suggested. It is not recommended to discontinue or adjust any AL-specific immunosuppressive therapy [50]

However, the landscape of current virus variants is changing rapidly, and several studies are evaluating the efficacy of the above-mentioned therapeutic options regarding new circulating virus variants. Therefore, the latest recommendations by the ECIL should be individually discussed in every patient. In general, patients with AL should be treated early with antiviral therapy to prevent disease progression. In addition, rapid elimination of the virus should be attempted, otherwise patients with underlying AL may remain positive for a very long time. When hyper-inflammatory syndrome occurs, it should be treated rapidly as well [50].

Another major challenge in AL patients who are infected with SARS-CoV-2 is the prevention of thromb-embolic events, since infection with SARS-CoV-2 is associated with a higher risk for developing thromb-embolic events [98]. The current ASH recommendations conclude that:patients without suspected/confirmed thrombosis who are treated in an outpatient setting should not receive anticoagulant thromboprophylaxis [99]patients without suspected or confirmed thrombosis who are acutely ill from COVID-19 should be treated with therapeutic-intensity anticoagulation. Acutely ill patients are defined as those who are administered to the hospital due to COVID-19 symptoms but who are not treated at an intensive care unit [99]patients without suspected or confirmed thrombosis who are critically ill from COVID-19 should be treated with prophylactic-intensity anticoagulation. Critically ill patients are defined as those who are administered to an intensive care unit due to COVID-19-associated symptoms [99]

Since the majority of AL patients suffer from thrombocytopenia due to the underlying disease or AL therapy, anticoagulation should be adjusted regarding the current thrombocyte count and coagulation status in each patient according to the local hospital standard operating procedures.

## 6. Conclusions

Most available data on the impact of COVID-19 on AL relate to the pre-vaccination era and include very scarce data on adult ALL. The COVID-19-associated mortality rate in adult AL patients is high (20–52%). Patients with AML display a particularly high COVID-19-related mortality rate. Therapy delay is recommended until SARS-CoV-2 negativity if feasible; therapy discontinuation is not recommended. Intensive treatment should be offered to every eligible patient. Since the impact of neutropenia and lymphocytopenia is controversial, no clear conclusion can be made on the impact on mortality. It is recommended to offer vaccination to all AL patients. High response rates around 80–96% could be observed. Post-vaccination anti-SARS-CoV-2 antibody titers should be obtained, and when negative or low, passive immunization with a suitable SARS-CoV-2 monoclonal antibody should follow. In general, patients with AL should be treated early with antiviral therapy to prevent disease progression and promote the rapid elimination of the virus. When hyper-inflammatory syndrome occurs, it should be treated rapidly. Further large registries are needed to provide further data and evidence-based recommendations for AL patients with concomitant SARS-CoV-2 infection. Due to the lack of current data on COVID-19 in AL patients after vaccination and the impact of the new virus variants on clinical courses, no changes to the recommendations can be derived in this regard.

## Figures and Tables

**Table 1 cancers-14-03711-t001:** Overview of the most important studies on clinical courses of COVID-19 in patients with acute leukemia (Part 1 including study design and number of study population).

Author (Year)	Observation Period	Number of AML Patients (*n*)	Number of ALL Patients (*n*)	Study Design	Reference
Marchesi et al. (2022)	02/2020–10/2021	388	NA	Multicenter, European registry	[31]
Pagano et al. (2021)	03–12/2020	497	169	Multicenter, Eurpean registry	[34]
García-Suárez et al. (2020)	02–05/2020	61	13	Multicenter, Spanish registry	[38]
Palanques-Pastor et al. (2021)	03–05/2020	108	NA	Observational study	[35]
Mitrovic et al. (2021)	Pre-vaccination era	51 ^1^	51 ^1^	Unknown	[32]
Yigenoglu et al. (2020)	03–06/2020	40	18	Retrospective, Turkish trial	[41]
Demichelis-Gómez et al. (2021)	02–07/2020	83 ^1^	83 ^1^	Multicenter, prospective Latin American cohort study	[33]
Martínez et al. (2020)	03–05/2020	117 ^1^	117 ^1^	Multicenter, Spanish observational study	[36]
Chiaretti et al. (2022)	02/2020–02/2021	NA	63	Multicenter, Italian observational study	[43]
Ribera et al. (2021)	02/2020–02/2021	NA	52	Multicenter, Spanish observation study	[44]
Wood et al. (2020)	04–07/2020	80 ^1^	80 ^1^	Multicenter, global registry	[42]

^1^ Acute leukemia patients in general without a division into acute myeloid leukemia or acute lymphoblastic leukemia.

**Table 2 cancers-14-03711-t002:** Overview of the most important studies on clinical courses of COVID-19 in patients with acute leukemia (Part 2 including COVID-19-related mortality and overall mortality).

Author (Year)	Proportion of Severe COVID-19 (%)	Proportion of Critical COVID-19 (%)	Death Caused Primary by COVID-19 (AML/ALL) (%)	Overall Mortality (AML/ALL) (%)	Reference
Marchesi et al. (2022)	41	21	20/NA	46/NA	[31]
Pagano et al. (2021)	NA	NA	NA	40/26	[34]
García-Suárez et al. (2020)	48 and 31 in AML and ALL, respectively	26 and 23 in AML and ALL, respectively	NA	44/15	[38]
Palanques-Pastor et al. (2021)	NA	NA	NA	44/NA	[35]
Mitrovic et al. (2021)	NA	NA	29 ^1^	NA	[32]
Yigenoglu et al. (2020)	NA	NA	NA	20/17	[41]
Demichelis-Gómez et al. (2021)	48	48	48/52	NA	[33]
Martínez et al. (2020)	54 (including severe and clinical courses)	NA	NA	48 ^1^	[36]
Chiaretti et al. (2022)	NA	NA	NA	NA/11	[43]
Ribera et al. (2021)	NA	NA	NA/29	NA/33	[44]
Wood et al. (2020)	63 (including moderate and severe courses)	NA	NA	33 ^1^	[42]

^1^ Acute leukemia patients in general without a division into acute myeloid leukemia or acute lymphoblastic leukemia.

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
