# Peer review of "COVID-19 and Adult Acute Leukemia: Our Knowledge in Progress"

_cancers, 2022, doi:10.3390/cancers14153711_

Round 1

Reviewer 1 Report

Interesting paper, raising an extremely important issue concerning SARS-CoV2 infection in patients with acute leukemias. Appropriate and up-to-date selection of reference.

I propost to elaborate on the use of monoclonal anti-SARS-CoV-2 antibodies in immunosuppressed patients and reference the literature data on preliminary results of studies on infection prevention and early exposure - p. 6-7, v. 305-318.

Author Response

“Interesting paper, raising an extremely important issue concerning SARS-CoV2 infection in patients with acute leukemias. Appropriate and up-to-date selection of reference.

I propost to elaborate on the use of monoclonal anti-SARS-CoV-2 antibodies in immunosuppressed patients and reference the literature data on preliminary results of studies on infection prevention and early exposure - p. 6-7, v. 305-318.”

Answer:

We thank the reviewer for this valuable feedback and appreciate the reviewers time and effort to improve our manuscript. As suggested, we elongated the monoclonal antibody section by providing preliminary information on bebtelovimab as the only neutralizing monoclonal antibody which showed efficacy against the Omicron variant (please refer to lines 310-316). However, to the best of our knowledge, the final results of the BLAZE-4 study have not been published yet.

Reviewer 2 Report

The study demonstrates that acute leukemia patients with SARS-CoV-2 infection may have high COVID-19-related mortality. Careful consideration is needed for the patients with acute leukemia.

Please find the below as the specific comments for the manuscript.

The research investigates the influence of adult acute leukemia on SARS-CoV-2 infection. I consider the topic relevant in the field, since the coronavirus infection and cancer may be closely associated. It adds the possibility that the SARS-CoV-2 infection affects on the acute leukemia prognosis. The references may be added for Table 1 and 2. Conclusion is consistent with the search results. There is a limitation in which the current data on COVID-19 in acute leukemia patients after vaccination is lacking. Some additional references regarding the vaccination may be added. A figure that indicates the scheme of the literature search may be added. The differences between Table 1 and Table 2 may be highlighted and described in the titles of the tables.

Author Response

“The study demonstrates that acute leukemia patients with SARS-CoV-2 infection may have high COVID-19-related mortality. Careful consideration is needed for the patients with acute leukemia.

Please find the below as the specific comments for the manuscript.

The research investigates the influence of adult acute leukemia on SARS-CoV-2 infection. I consider the topic relevant in the field, since the coronavirus infection and cancer may be closely associated. It adds the possibility that the SARS-CoV-2 infection affects on the acute leukemia prognosis. The references may be added for Table 1 and 2.”

Answer: We thank the reviewer for this feedback. Indeed, all references regarding tables 1 and 2 are already added (please refer to tables 1 and 2, last column).  

“Conclusion is consistent with the search results. There is a limitation in which the current data on COVID-19 in acute leukemia patients after vaccination is lacking. Some additional references regarding the vaccination may be added.”

Answer: We thank the reviewer for this valuable feedback.  To the best of our knowledge, we included the most important references regarding the efficacy of COVID-19 vaccination in patients with acute leukemia in our review. However, since we have not performed a systematic search according to the PRISMA guidelines, there is a residual risk of single references missing in our review.

“A figure that indicates the scheme of the literature search may be added. The differences between Table 1 and Table 2 may be highlighted and described in the titles of the tables.”

Answer: Again, we appreciate the reviewers’ suggestions. We agree with the reviewer that in the case of a systematic review, a figure can help to provide an overview of the literature research method. However, since our review here is not based on a systematic search according to the PRISMA guidelines, we cannot present a figure to provide a systematic overview of our search strategy. To highlight the differences between table 1 and table 2, we precised both tables’ titles as suggested.
